# Integrated In Silico Analyses Identify PUF60 and SF3A3 as New Spliceosome-Related Breast Cancer RNA-Binding Proteins

**DOI:** 10.3390/biology11040481

**Published:** 2022-03-22

**Authors:** Jennyfer M. García-Cárdenas, Isaac Armendáriz-Castillo, Andy Pérez-Villa, Alberto Indacochea, Andrea Jácome-Alvarado, Andrés López-Cortés, Santiago Guerrero

**Affiliations:** 1Escuela de Medicina, Facultad de Ciencias Médicas de la Salud y de la Vida, Universidad Internacional del Ecuador, Quito 170113, Ecuador; jennyfergrc7@gmail.com (J.M.G.-C.); khauranshy@gmail.com (A.J.-A.); 2Facultade de Ciencias, Universidade da Coruña, 15071 A Coruna, Spain; 3Latin American Network for the Implementation and Validation of Clinical Pharmacogenomics Guidelines (RELIVAF-CYTED), 28001 Madrid, Spain; isaac.arcas@gmail.com (I.A.-C.); andypzvi@gmail.com (A.P.-V.); 4Instituto Nacional de Investigación en Salud Pública, Quito 170136, Ecuador; 5Facultad de Ingenierías y Ciencias Aplicadas, Universidad Internacional SEK, Quito 170302, Ecuador; 6Centre for Genomic Regulation (CRG), Barcelona Institute of Science and Technology, 08003 Barcelona, Spain; aindacocheac@gmail.com; 7Programa de Investigación en Salud Global, Facultad de Ciencias de la Salud, Universidad Internacional SEK, Quito 170302, Ecuador; 8Facultad de Medicina, Universidad de Las Américas, Quito 170124, Ecuador

**Keywords:** RBPs, breast cancer, cancer driver genes, in silico analysis

## Abstract

**Simple Summary:**

Globally, breast cancer (BC) is the most common cancer in women. Although numerous studies have attempted to address this worldwide health problem, it has not yet been possible to understand cancer in its entirety, mainly because most of the investigations have been focused on traditional molecular traits of DNA. Thus, new characteristics of breast tumorigenesis must be tackled, such as RNA-binding proteins (RBPs), which are crucial regulators of important cellular processes. To identify novel breast cancer RNA-binding proteins, we integrated several bioinformatic resources derived from experimentation on BC patient samples and cell lines. Consequently, we identified five putative breast cancer RNA-binding proteins (PUF60, TFRC, KPNB1, NSF, and SF3A3) showing strong tumorigenic characteristics. Supplementary investigation of the molecular and cellular functions of these proteins identified PUF60 and SF3A3 as new spliceosome-related breast cancer RNA-binding proteins. Further experimentation should center on these five RBPs to identify their role in breast tumorigenesis and potentially discover new druggable targets.

**Abstract:**

More women are diagnosed with breast cancer (BC) than any other type of cancer. Although large-scale efforts have completely redefined cancer, a cure remains unattainable. In that respect, new molecular functions of the cell should be investigated, such as post-transcriptional regulation. RNA-binding proteins (RBPs) are emerging as critical post-transcriptional modulators of tumorigenesis, but only a few have clear roles in BC. To recognize new putative breast cancer RNA-binding proteins, we performed integrated in silico analyses of all human RBPs (*n* = 1392) in three major cancer databases and identified five putative BC RBPs (PUF60, TFRC, KPNB1, NSF, and SF3A3), which showed robust oncogenic features related to their genomic alterations, immunohistochemical changes, high interconnectivity with cancer driver genes (CDGs), and tumor vulnerabilities. Interestingly, some of these RBPs have never been studied in BC, but their oncogenic functions have been described in other cancer types. Subsequent analyses revealed PUF60 and SF3A3 as central elements of a spliceosome-related cluster involving RBPs and CDGs. Further research should focus on the mechanisms by which these proteins could promote breast tumorigenesis, with the potential to reveal new therapeutic pathways along with novel drug-development strategies.

## 1. Introduction

Breast cancer (BC) is the leading cause of cancer-associated death (15%: 626,679 cases) and the most commonly diagnosed cancer (24%: 2,088,849 cases) among women worldwide [1]. BC is characterized by a complex interaction between environmental factors and biological traits, such as gene deregulation, hormone disruption, or ethnicity [2,3,4]. Despite treatment efforts, advanced BC with distant organ metastasis is considered to be incurable [2]. Therefore, a better understanding of BC’s molecular processes is still pertinent to identifying new therapeutic targets. Current oncological research generates large-scale datasets that harbor essential aspects of tumor biology. For instance, the Cancer Genome Atlas (TCGA), with over 2.5 petabytes of data, has molecularly characterized over 20,000 patient samples covering 33 cancer types [5,6,7,8,9,10]. Additionally, the Cancer Dependency Map (DepMap) project, using loss-of-function genetic screens, has identified essential genes for cancer proliferation and survival ex vivo [11,12,13]. Additionally, the Human Protein Atlas (HPA) constitutes a comprehensive resource to explore the human proteome in healthy and tumoral human tissues [14,15,16]. Although these datasets have completely redefined cancer drug development, diagnosis, and treatment, additional fundamental features of oncogenesis, tumor growth, and dissemination remain to be discovered. In this respect, post-transcriptional regulation of tumorigenesis represents an understudied trait of cancer research [17].

RNA-binding proteins (RBPs) are particularly relevant due to their implication in every post-transcriptional step of gene expression: RNA splicing, transport, stability, translation, and localization. As a result, genomic alterations of these proteins lead to dysfunctional cellular processes, but only a few have defined functions in BC [18,19,20,21,22,23,24,25]. To date, 1393 RBPs have been experimentally identified in the human RNA interactome [26]. Despite efforts to understand their role in cancer [27,28], an integrated analysis of the aforementioned databases along with other in silico approaches is still missing for BC. To shed light on this matter, we analyzed and integrated RBPs genomic alterations, protein–protein interaction (PPI) networks, immunohistochemical profiles, and loss-of-function experiments to find new putative breast cancer RNA-binding proteins.

## 2. Materials and Methods

### 2.1. Gene Sets

A total of 1393 RBPs were extracted from Hentze et al. [26] and checked for new annotations using Ensembl (http://www.ensembl.org (accessed on 5 February 2022)) [29,30]. Only one duplicate was found: ENSG00000100101 and ENSG00000273899, both correspond to *NOL12*, leaving a final list of 1392 RBPs. BC genes (*n* = 171) were obtained from the Network of Cancer Genes 6.0 (NCG6) [31]. Non-cancer gene list was constructed from Piazza et al. [32], without RBPs and NCG6 [31] genes, and reanalyzed using Piazza’s OncoScore algorithm (https://www.galseq.com/next-generation-sequencing/oncoscore-software (accessed on 7 January 2022)), giving a final list of 177 non-cancer genes (Appendix A).

### 2.2. Genomic Analysis

Genomic alterations of RBPs, non-cancer, and BC genes were analyzed through the cBioPortal (https://www.cbioportal.org (accessed on 12 March 2022)) [33,34] using the Breast Invasive Carcinoma (TCGA, PanCancer Atlas) database (*n* = 994 complete samples) and the Clinical Proteomic Tumor Analysis Consortium (CPTAC) database (*n* = 122 complete samples) [5,6,7,8,9,10,35]. To compare the aforementioned gene sets, genomic alterations per protein were corrected by the number of genes or individuals. A Mann–Whitney U test was used to compare genomic alterations between gene sets or clinical characteristics (Appendix A).

### 2.3. Network Construction

Experimental and database interactions (Appendix A) between RBPs (*n* = 1392) and BC proteins (*n* = 171) [31], having an interaction score of 0.9 (highest confidence), were extracted from the STRING database (Appendix A) [36] and visualized using the Cytoscape 3.7.1 (Seattle, USA) platform [37].

### 2.4. Protein Expression Analysis

Immunohistological levels of 1212 available RBPs in normal and BC tissues were extracted from Protein Atlas version 18.1 (https://www.proteinatlas.org (accessed on 15 June 2021)) [14,15,16]. Expression levels of normal tissues were taken from glandular cells, while a consensus level was manually generated for BC tissues (Appendix A) based on tissue level frequency. Immunohistological images were taken from https://www.proteinatlas.org/ENSG00000182481-KPNA2/tissue/breast#img (accessed on 15 June 2021) (KPNA2 staining of normal tissue), https://www.proteinatlas.org/ENSG00000182481-KPNA2/pathology/tissue/breast+cancer#img (accessed on 15 June 2021) (KPNA2 staining in tumoral tissue), https://www.proteinatlas.org/ENSG00000138757-G3BP2/tissue/breast#img (accessed on 15 June 2021) (G3BP2 staining in normal tissue), https://www.proteinatlas.org/ENSG00000138757-G3BP2/pathology/breast+cancer#img (accessed on 15 June 2021) (G3BP2 staining in tumoral tissue), https://www.proteinatlas.org/ENSG00000109111-SUPT6H/tissue/breast#img (accessed on 15 June 2021) (SUPT6H staining in normal tissue), and https://www.proteinatlas.org/ENSG00000138757-G3BP2/pathology/breast+cancer#img (accessed on 15 June 2021) (SUPT6H staining in tumoral tissue).

### 2.5. Cancer-Dependency Analysis

RBP cancer-dependency scores from CERES [11] (1288 available RBPs) and DEMETER2 [12,13] (1290 available RBPs) were obtained from the Dependency Map (DepMap) portal (https://depmap.org/portal (accessed on 10 June 2021)). Molecular subtypes of 82 (DEMETER2 [12,13]) and 28 (CERES [11]) BC cell lines were obtained from Smith et al. [38], Dai et al. [39], and Kao et al. [40] (Appendix A).

### 2.6. Cancer-Related Networking Analysis

Previously prioritized RBPs (PUF60, TFRC, KPNB1, NSF, and SF3A3) were integrated into a disease gene network (filtered by RBPs [*n* = 125] and CDGs ((*n* = 202 genes)) by using the HumanNet XN (fully extended functional gene network) v2 software (https://www.inetbio.org/humannet (accessed on 9 July 2021)) and visualized through Cytoscape V3.8.2 [37]. We then used MCODE [41] to find complexes within the network according to level-3 parameters: node score cutoff = 0.1, fluff = 0, and no haircut. The resulting network was interpreted through CORUM [42], a database of mammalian protein complexes (https://mips.helmholtz-muenchen.de/corum (accessed on 8 July 2021)).

## 3. Results

### 3.1. An Overview of RNA-Binding Protein Genomic Alterations in Breast Cancer

To globally assess the potential role of RBPs in BC, we performed complementary analyses, which are depicted in Figure 1.

Then, to evaluate the potential role of RBPs in BC versus well-known BC genes, we interrogated the Breast Invasive Carcinoma (TCGA, PanCancer Atlas) and Breast Cancer (CPTAC) [5,6,7,8,9,10,35] database for genomic alterations of RBPs (*n* = 1392), BC genes (*n* = 171) [31], and non-cancer genes (*n* = 170) [32] (Table 1). As shown in Figure 2A, both genomic alteration frequencies of RBPs and BC genes were significantly higher than the ones observed for non-cancer genes. Interestingly, RBPs present a similar degree of genomic alterations as BC genes (Figure 2A), highlighting the putative role of RPBs in BC.

To obtain insights into how these proteins are altered in BC, we cataloged their genomic alteration types. As shown in Figure 2B and (Appendix A), most genomic alterations are related to an overrepresentation of the mRNA (68.7%) or gene loci (15.4%).

### 3.2. Identification of Highly Altered Breast Cancer RNA-Binding Proteins

To identify breast cancer-related RNA-binding proteins, we next interrogated the Network of Cancer Genes 6.0 (NCG6) [31] for RBP having known or predicted cancer driver roles. NCG6 harbors the most recent catalog of cancer driver genes (CDG) [31]. Thus, we identified 225 RBPs, 14 implicated in BC (2 oncogenes, 4 tumor suppressors, and 8 unknown), indicating that these proteins remain poorly studied in breast carcinogenesis, and 211 related to other cancer types (21 oncogenes, 24 tumor suppressors, and 166 unknown) (Figure 3A, Appendix A).

To categorize putative RBPs implicated in tumor progression or suppression, we analyzed RBPs’ genomic alterations based on their progressor or suppressor profiles. Tumor progressors tend to be overexpressed (mRNA upregulation or genomic amplification), while suppressors are downregulated (mRNA downregulation or genomic deletion) in malignant cells [57]. Gene mutations or fusions have been observed in both tumor progressors and suppressors. On this basis, we identified highly altered breast cancer RNA-binding proteins (Table 1 and Appendix A). Interestingly, 30% of all human RBPs interact with the tumor suppressor ESR2 (Figure 3B) [43]. We also found known BC progressor and suppressor proteins, such as DAP3 [18], MTDH [19], or CCAR2 [20], which validate our strategy (Table 1 and Appendix A). This analysis also reveals proteins that have not been related to tumorigenesis, and yet they are highly altered in BC (e.g., TFB2M, C1ORF131, or DDX19A) (Table 1 and Appendix A).

To further identify important RBPs implicated in BC, we analyzed RBPs’ genomic alterations by subtype (Normal, LumA, LumB, Her2, and Basal) (Appendix A) or staging (Stage I to IV) (Appendix A). As shown in Figure 3C, RBP genomic alterations found in the Basal subtype samples were statistically significant compared to other subtypes (*p* < 0.001). Similarly, RBP genomic alterations of Stage IV samples were statistically significant compared to other stages (*p* < 0.001) (Figure 3D). Individually, some RBPs reached high frequencies of genomic alterations per subtype (Figure 3C, Appendix A) or stage (Figure 3D, Appendix A). For instance, ARF1, the most altered protein in Stage IV (Figure 3D), has been shown to promote BC metastasis [58]; PARP1 has also been demonstrated to enhance metastasis not only in BC [59] but also in other cancer types [60]. In contrast, SCAMP3 and HEATR6, which present similar degrees of genomic alterations (Figure 3D), have not been studied in BC.

### 3.3. RNA-Binding Proteins Interact with Well-Known Breast Cancer Proteins

Networking analysis has proved useful in identifying RNA regulons and crucial tumoral proteins [57]. On this basis, we next explored PPIs between RBPs (*n* = 1392) and well-known BC proteins (*n* = 171) [31] using the STRING database [61]. The interactions were obtained from experiments and databases; the interaction score was 0.9. This is the highest possible confidence of an interaction to be true based on all the available evidence. Thus, we identified 113 BC proteins interacting with 398 RBPs (Appendix A). By narrowing down our analysis to experimental interactions only (Figure 4), we observed two main networks around SF3B1 and CDC5L proteins. According to the g:Profiler [62], proteins interacting with SF3B1 are implicated in RNA splicing (P_adj_ = 3.783 × 10^−34^; GO:0000377) (*p*-value adjusted (P_adj_) for multiple testing using the Benjamin–Hochberg method), while proteins connected to CDC5L are mainly involved in chromatin binding (P_adj_ = 1.500 × 10^−2^; GO:0003682). We also observed proteins with both BC and RNA-binding features present in the two main networks: SF3B1, CTNNA1, RBMX, and SPEN. Additionally, 18 RBPs interact with at least 1 BC protein. Thus, we identified RBPs that may have a putative role in BC’s molecular pathways through PPIs.

### 3.4. Identification of Differentially Expressed RNA-Binding Proteins in Breast Tumor Tissues

The Human Protein Atlas (HPA) constitutes [14,15,16] a major effort to address protein expression in healthy and tumoral human tissues. We, therefore, identified RBPs with a different protein expression profile in tumor breast tissues. To this end, we compared immunohistochemical levels (not detected, low, medium, and high) of 1212 available RBPs between normal and cancerous breast tissues (Figure 5A, Appendix A). Most RBPs presented common immunohistochemical levels between both breast tissues: not detected (*n* = 130), low (*n* = 52), medium (*n* = 366), and high (*n* = 72) (Figure 5A). Moderate protein expression changes, defined by one level variation (e.g., not detected to low or medium to high), were observed in 406 RBPs.

To identify RBPs with highly altered protein expression profiles in tumor tissues, we categorized RBPs with a twofold variation level as upregulated or downregulated compared with normal tissues; thus, we identified 24 upregulated and 62 downregulated RBPs (Figure 5A, Appendix A). As expected, our approach revealed well-known BC proteins, such as KPNA2 [21] or G3BP2 [22], which validate our analysis. KPNA2 is highly expressed in BC tissues (7 out of 12 tumor samples are classified as high) (Figure 5B, Appendix A). On the contrary, G3BP2 expression is reduced in tumoral breast tissues (Figure 5B, Appendix A). We also observed two RBPs that have never been studied in BC, DARS2 (overexpressed) and SUPT6H (downregulated) (Figure 5B, Appendix A).

### 3.5. Exploring RNA-Binding Proteins Breast Cancer Dependencies

Most RBPs present numerous genomic alterations (Figure 2 and Figure 3C,D; Appendix A), making it difficult to detect essential RBPs for cell proliferation and/or survival, i.e., breast cancer RBPs dependencies. Thus, we analyzed 1288 available RBPs on CERES [11] and 1290 available RBPs on DEMETER2 [12,13] through the DepMap portal (https://depmap.org/portal (accessed on 20 June 2021)). Both initiatives report loss-of-function screens performed in several human cancer cell lines [11,12,13].

Figure 6A shows the distribution of dependency scores of all available RBPs in 82 (DEMETER2 [12,13]) and 28 (CERES [11]) BC cell lines. The dependency score expresses how vital a gene is in a target cell line; if the score is greater than 0.5, the cell line is considered dependent. The genome-scale RNAi loss-of-function screens (DEMETER2 [12,13]) identified 90 essential RBPs (Figure 6A), being SNRPD1, SF3B1, SF3B2, RPL5, ARCN1, EIF3B, RAN, COPB1, RPL14, and VCP (mean dependency scores ranging from −1.3 to −1.5) the top ten essential RBPs for BC survival (Appendix A). On the other hand, genome-scale CRISPR-Cas9 loss-of-function screens (CERES [11]) determined 176 essential RBPs (Figure 6A), being RAN, HSPE1, SNRNP200, SNRPD1, SARS, EEF2, RPL37, CCT3, KPNB1, and RPL23 (mean dependency scores ranging from −1.5 to −1.8) the top ten essential RBPs for tumor survival (Appendix A). In toto, 207 essential RBPs were identified by both computational methods (Figure 6A; Appendix A).

To identify essential RBPs per BC molecular subtype, we first updated subtypes by merging data from Smith et al. [38], Dai et al. [39], and Kao et al. [40] (Appendix A). We next identified and compared 203 LumA, 96 LumB, 206 Her2, and 212 Basal essential RBPs (Figure 6B; Appendix A). Thus, we identified essential RBPs for each BC subtype: seven LumA (HSPD1, UBE2M, SART3, USP36, GTPBP4, DHX33, and UPF1), five LumB (RPS21, GNL3L, ZNF207, AQR, and RPL17-C18orf32), seven Her2 (DDX39B, NMT1, ISY1, DARS, HEATR1, MAT2A, and SYF2), and nine Basal (EIF3C, UTP20, TXN, NOP58, ALDOA, CCT2, NOP2, DDX54, and PRMT1) (Figure 6B).

### 3.6. Unraveling Putative Breast Cancer RNA-Binding Protein

Cancer-related RBPs control hundreds of tumor mRNAs, interact with well-known cancer driver proteins, and appear to be highly altered in cancer genomic databases and tumor tissues [57]. Therefore, we reasoned that the integration of our previous analyses could narrow down the identification of a potential breast cancer RNA-binding protein.

To this end, we focused on RBPs with putative tumor progression profiles. Thus, we overlapped our previous results as follows: (1) 348 RBPs belonging to the first quartile of most genomically altered RBPs concerning tumor-progression-related alterations (mRNA upregulation, genomic amplification, gene mutations, or fusions); (2) all 398 RBPs presenting PPIs with well-known BC proteins (Appendix A); (3) 160 RBPs with at least one immunohistochemical variation level towards protein overexpression (e.g., not detected to low); (4) all 207 essential BC RBPs (Figure 6A; Appendix A).

We found five RBPs presenting the aforementioned tumor-associated characteristics, TFRC, KPNB1, PUF60, NSF, and SF3A3 (Figure 7). TFRC and KPNB1 have been previously implicated in BC [63,64,65,66], while PUF60 has been associated with colon and non-small cell lung cancer [67,68]. Interestingly, NSF and SF3A3 have never been studied in cancer. We also found 14 RBPs showing high genomic alterations, PPIs with BC proteins, and altered protein expression profiles in tumoral tissues. Although these proteins are not needed for tumor survival ex vivo (Appendix A), they could be implicated in other tumoral processes; indeed, 11 of these RBPs have been described as BC tumor progressors [18,69,70,71,72,73,74,75,76,77,78,79]. Interestingly, PLEC has not been related to BC but promotes the migration and invasion of neck squamous cell carcinoma [80]. In addition, PRPF3 and MAGOHB have not been linked to cancer before. In fact, PRPF3 alterations have been related to Retinitis pigmentosa and MAGOHB to Metaphyseal Chondrodysplasia, Schmid Type, and Hermansky–Pudlak Syndrome 3.

### 3.7. PUF60 and SF3A3 Are Central Elements of a Spliceosome-Related Network Involving RNA-Binding Proteins and Cancer Driver Genes

To better understand the cellular functions of these prioritized RBPs (TFRC, KPNB1, PUF60, NSF, and SF3A3) in cancer, we next interrogated the HumanNet v2 [81,82]. This tool allowed us to integrate these five RBPs into a disease gene network. We first obtained an initial network of 2231 interactions (Appendix A). To narrow down the analysis to cancer-relevant interactions, we then filtered the network by CDGs (*n* = 202 genes) and RBPs (*n* = 125 genes) and used MCODE [41] to find protein complexes within the network; the largest one and more relevant was formed by 36 nodes and 591 edges. The CORUM [42] database identified 34 of these 36 proteins as a part of the spliceosome complex where PUF60 and SF3A3 are central elements interacting with several RBPs and the cancer driver gene (Figure 8).

## 4. Discussion

Current oncological research generates large-scale datasets that contain undiscovered strategic features of molecular mechanisms underlying the growth and metastasis of tumors, and yet these databases are not fully exploited. Integrated in silico analyses of these data could therefore lead to the discovery of new cancer proteins.

We first revealed that RBPs are equally altered as well-known BC proteins (Figure 2A); this was expected since many RBPs are highly altered across cancer types [28] and have been linked in silico to cancer-related cellular processes [83]. We found that most RBPs’ genomic alterations in BC are mRNA upregulation (68.7%) and amplification (15.4%) (Figure 2B). This probably will increase RBPs’ cellular concentrations, leading to dysfunctional post-transcriptional processes.

To determine how many RBPs have been previously studied in BC, we analyzed the most recent catalog of CDG, NCG6 [31]. Only 14 RBPs were cataloged as BC driver genes (Figure 3A). This indicates that RBPs have been poorly investigated in breast carcinogenesis. Thus, to identify new putative breast cancer RNA-binding proteins, we first explored their genomic alteration profiles associated with tumor progression or suppression (Table 1 and Appendix A). As expected, we identified well-known BC-progressor and -suppressor proteins, such as DAP3 [18], MTDH [19], or CCAR2 [20], which validate our strategy (Table 1). On the contrary, our strategy revealed RBPs that have not been associated with tumorigenesis, and yet they are highly altered in BC (e.g., TFB2M, C1ORF131, or DDX19A) (Table 1). Interestingly, the most altered RBP in our analysis, MRPL13, has never been studied in cancer. MRPL13, along with other highly altered RBPs (Table 1), has only been shown to interact with ESR2, a tumor suppressor in breast and other cancer types [43]. This observation led us to investigate how many RBPs interact with ESR2; strikingly, we found that 30% of all RBPs interact with this receptor (Figure 3B) [43]. ESR2 could probably exert its suppressive activity through post-transcriptional mechanisms involving several RBPs; nevertheless, more research is needed to understand this observation.

Second, to further characterize RBPs associated with BC subtypes and staging, we analyzed RBPs’ genomic alterations (Figure 3C,D). Interestingly, RBPs’ genomic alterations gradually increased from the Normal to Basal subtype (Figure 3C), i.e., from a low to high proliferation stage [2]. Concordantly, metastasized tumors (Stage IV) showed high frequencies of RBPs’ genomic alterations compared to non-metastasized samples (Stage I to III) (Figure 3D). It seems, therefore, that RBPs are acting as BC progressors rather than suppressors, which agrees with their genomic-alteration profiles (Figure 2B). This analysis also revealed highly altered RBP per subtype or staging (Figure 3C,D; Appendix A), which could lead to the discovery of new clinical biomarkers or therapeutic targets. Indeed, SCAMP3 and HEATR6, which have not been studied in BC, presented similar degrees of genomic alterations (Figure 3D) compared to well-known metastasis drivers, ARF1 [58] and PARP1 [59]. In hepatocellular carcinoma cells, SCAMP3 knockdown has been shown to suppress cell proliferation [84], while HEATR6 has never been associated with tumorigenesis. Thus, more research is needed to understand their role in BC.

Interaction networks are useful for identifying crucial tumoral proteins [57]. In this regard, by analyzing PPIs between RBP and well-known BC proteins, we identified SF3B1 and CDC5L at the core of two main networks (Figure 4). While SF3B1 has been previously implicated in BC [85], CDC5L, which interacts with 14 BC proteins, has not been studied in this malignancy. However, CDC5L has been related to other cancer types, such as osteosarcoma [86] and prostate cancer [87].

We next exploited the HPA database [14,15,16] to identify differentially expressed RBPs in tumor breast tissues. We found 24 upregulated and 62 downregulated RBPs compared with normal tissues. Unsurprisingly, our analyses revealed RBPs that were already related to breast cancer. For instance, KPNA2, which has been known to enhance BC metastasis ex vivo [21], is highly expressed in BC tissues (7 out of 12 tumor samples are classified as high) (Figure 5B, Appendix A). On the contrary, G3BP2 expression is reduced in tumoral breast tissues (Figure 5B, Appendix A); accordingly, the loss of G3BP2 enhances tumor invasion and metastasis in vivo [22]. Interestingly, DARS2, which has never been related to BC, is upregulated in our analysis (10 out 12 tumor samples are classified as high) (Figure 5B, Appendix A) and has been associated with hepatocarcinogenesis [88], demonstrating its putative implication in BC. In addition, SUPT6H protein expression is diminished in breast tumoral tissues (Figure 5B, Appendix A) and has not been linked to this malignancy. Furthermore, SUPT6H knockdown is associated with DNA damage via the formation of RNA: DNA hybrids (R-loops) in HeLa cells [89], showing its possible role in breast tumorigenesis.

To identify essential RBPs for tumor survival, we next analyzed ex vivo loss-of-function screens, CERES [11] and DEMETER2 [12,13]. In toto, we identified 207 essential RBPs for tumor survival. This was expected since RBPs control every trait of RNA metabolism. However, only 59 were characterized as essential by both computational methods (Figure 6A; Appendix A). Although CERES [11] and DEMETER2 [12,13] did not test all human RBPs, future therapeutic post-transcriptional BC research could be focused on these 59 RBPs. However, more investigation is needed to deeply understand their carcinogenic roles. We also revealed essential RBPs per BC molecular subtype (Figure 6B) that could be analyzed to better understand subtype-related post-transcriptional processes.

In extending the scope of our previous analyses, we finally reasoned that the integration of all the databases examined could narrow down the identification of potential breast cancer RNA-binding proteins. As discussed before and depicted in Figure 2B and Figure 3C,D, RBPs seem to act as cancer progressors rather than suppressors. Thus, we focused on RBPs with putative tumor progression profiles and distinguished 19 RBPs with tumorigenic characteristics according to our analyses (Figure 7). As expected, most of them (13 out 19) have been described as BC tumor progressors, controlling different cellular processes such as migration, invasion, and metastasis. Interestingly, NSF, SF3A3, PRPF3, and MAGOHB have never been studied in cancer. While on the other hand, PUF60 has been associated with colon and non-small cell lung cancer [67,68], and PLEC has been shown to promote the migration and invasion of neck squamous cell carcinoma [80].

As depicted in Figure 7, we prioritized 5 RBPs according to our previous analyses. These putative BC progressor RBPs (PUF60, TFRC, KPNB1, NSF, and SF3A3) were integrated into a disease gene network to shed light on their molecular and cellular functions in cancer (Figure 8). Thus, we obtained a very intricate network of 2231 interactions (Appendix A), which emphasized the robust and complex network formed between RBP–RBP, RBP–CDG, and CDG–CDG. In addition to this complexity, some of these RBPs are also CDGs. Quattrone and Dassi already established that the RBP network is a hierarchical structure that is formed by clusters and chains that cooperate and compete on common target mRNAs controlling different cellular processes (e.g., splicing) [90]. This is also observed in our densely interconnected network, where PUF60 and SF3A3 are central elements of a spliceosome-related cluster involving RBPs and CDGs.

## 5. Conclusions

In sum, individual and integrated analysis of the aforementioned databases led us to identify RBPs that have never been studied in BC but displayed defined tumorigenic functions in other cancer types. Thus, based on their tumorigenic characteristics presented in this study and their roles in other cancer types, we identified five new putative breast cancer RBPs: PUF60, TFRC, KPNB1, NSF, and SF3A3. However, further research should focus on the mechanisms by which these proteins promote breast tumorigenesis, which holds the potential to discover new therapeutic pathways along with novel drug-development strategies.

## Figures and Tables

**Figure 1 biology-11-00481-f001:**
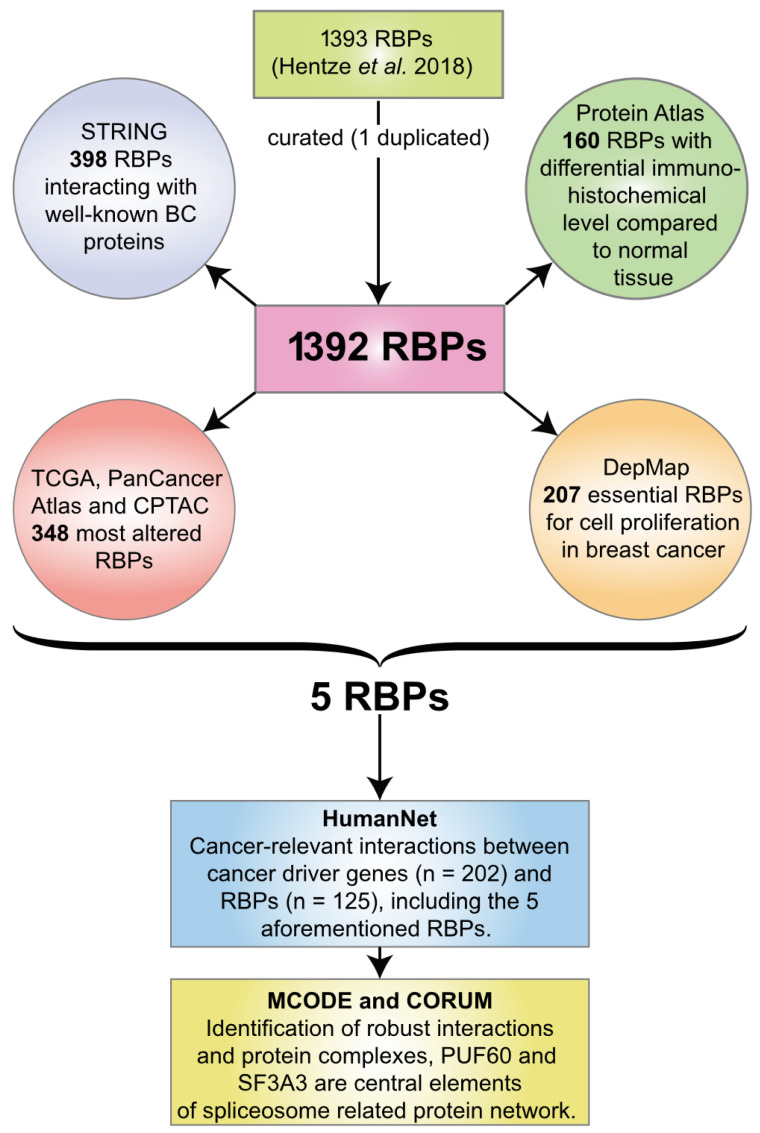
Workflow of the prioritization strategy. This scheme describes all major steps performed to identify PUF60 and SF3A3 as new spliceosome-related breast cancer RNA-binding proteins.

**Figure 2 biology-11-00481-f002:**
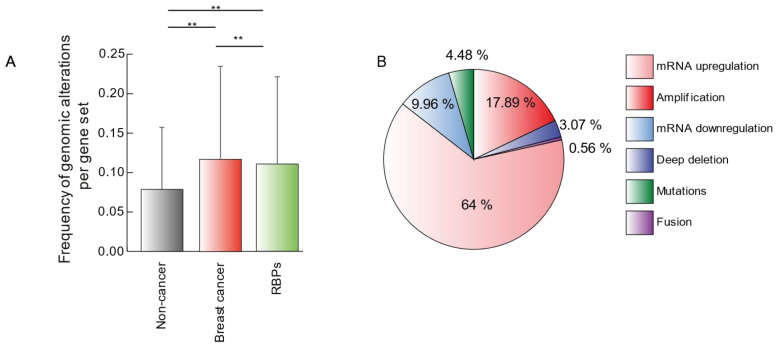
Genomic alterations of RBPs in BC. (**A**) Frequency of genomic alterations per gene set (non-cancer genes (*n* = 170), BC genes (*n* = 171), and RBPs [*n* = 1392]) using the Breast Invasive Carcinoma (TCGA, PanCancer Atlas) and Breast Cancer (CPTAC) database [5,6,7,8,9,10,35]. Genomic alterations per patient were corrected by the number of genes; a Mann–Whitney U test was used to compare genomic alterations between gene sets. ** = very significant difference; *ns* = not significant. (**B**) A pie chart describing RBPs’ genomic alteration types.

**Figure 3 biology-11-00481-f003:**
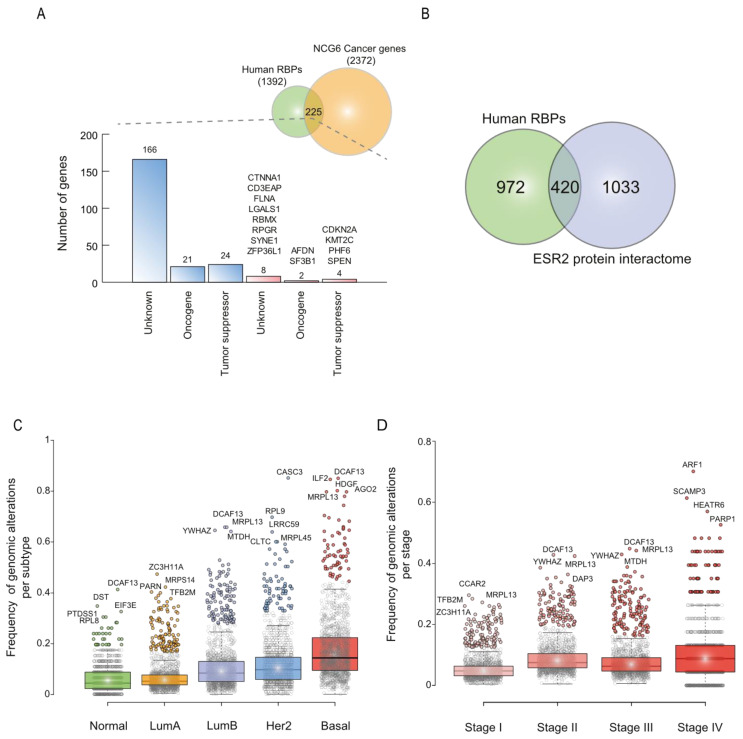
Identification of highly altered breast cancer RNA-binding proteins. (**A**) A histogram describing the status of RBPs in the Network of Cancer Genes 6.0 (NCG6). In blue, RBP status in other cancer types; in red, breast cancer RBPs. (**B**) A Venn diagram depicting the relationship between RBPs and ESR2 protein interactomes. RBPs genomic alterations per subtype (**C**) and stage (**D**), using the Breast Invasive Carcinoma (TCGA, PanCancer Atlas) and Breast Cancer (CPTAC) [5,6,7,8,9,10,35] database, are displayed. Genomic alterations per subtype and per stage were corrected by the number of patients; a Mann–Whitney U test was used to compare genomic alterations between sets. All possible comparisons between sets present significant differences (*p* < 0.001) except Normal vs. Lum A subtypes; *ns* = not significant.

**Figure 4 biology-11-00481-f004:**
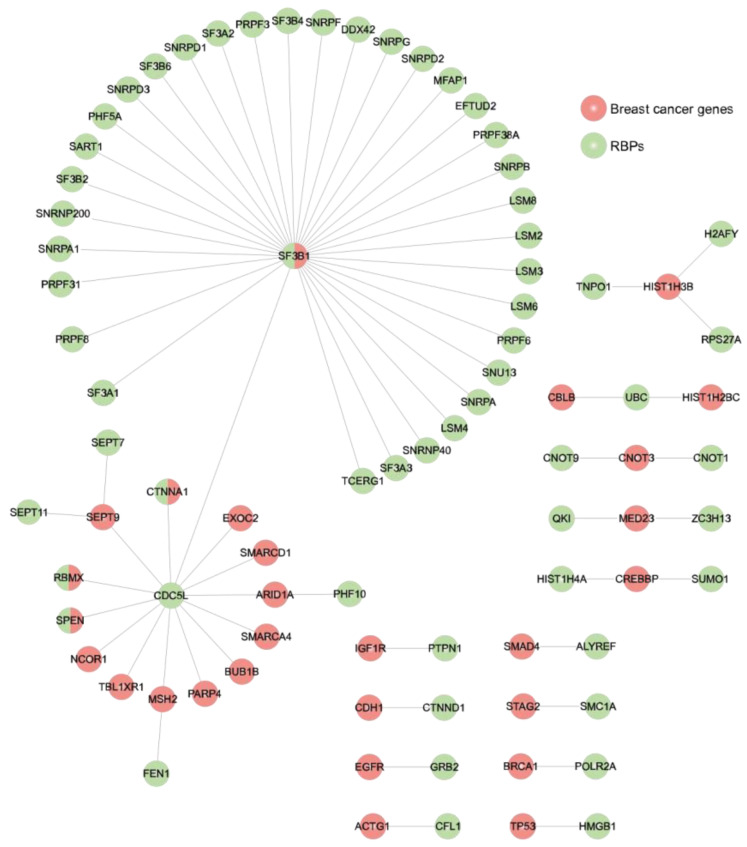
Experimental protein–protein interactions between RNA-binding proteins and well-known breast cancer proteins. An interaction network, constructed using STRING database and the Cytoscape 3.7.1 platform, is presented: red, BC proteins; green, RBPs.

**Figure 5 biology-11-00481-f005:**
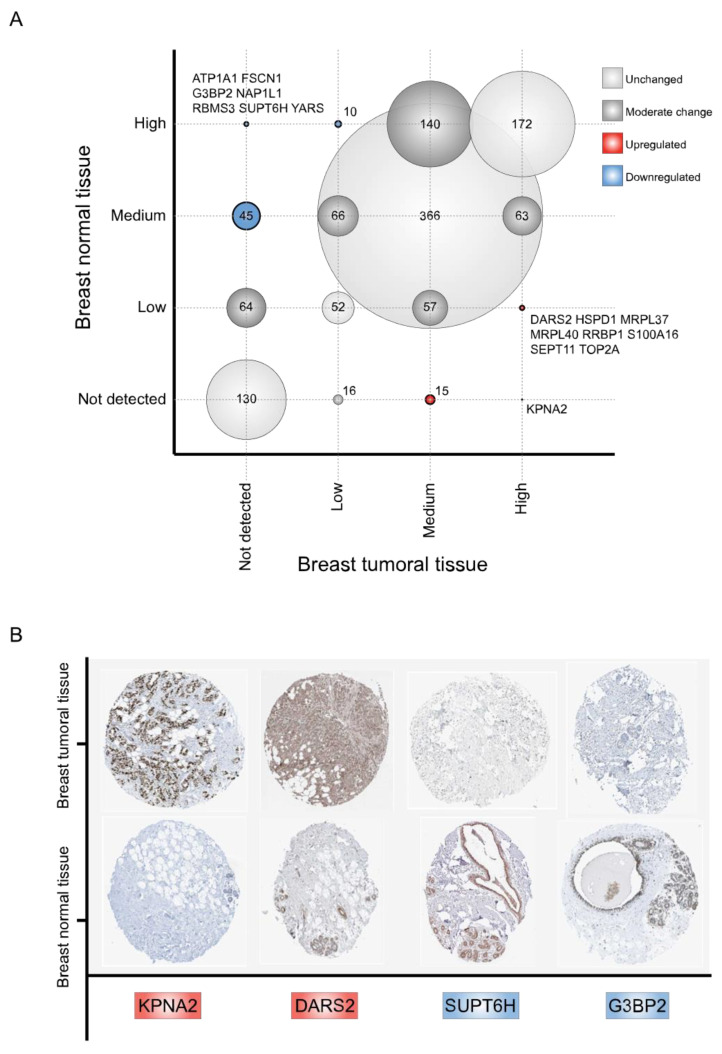
Immunohistochemical protein expression profile of RNA-binding proteins between healthy and tumor breast tissues. (**A**) A correlation plot, comparing RBPs immunohistochemical levels between normal and BC tissues, is presented. Circle sizes correlate with the number of RBPs in each intersection. (**B**) Representative immunohistochemical stains of four RBPS (upregulated: KPNA2 and DARS2; downregulated: G3BP2 and SUPT6H) on normal and tumor breast tissues according to the HPA.

**Figure 6 biology-11-00481-f006:**
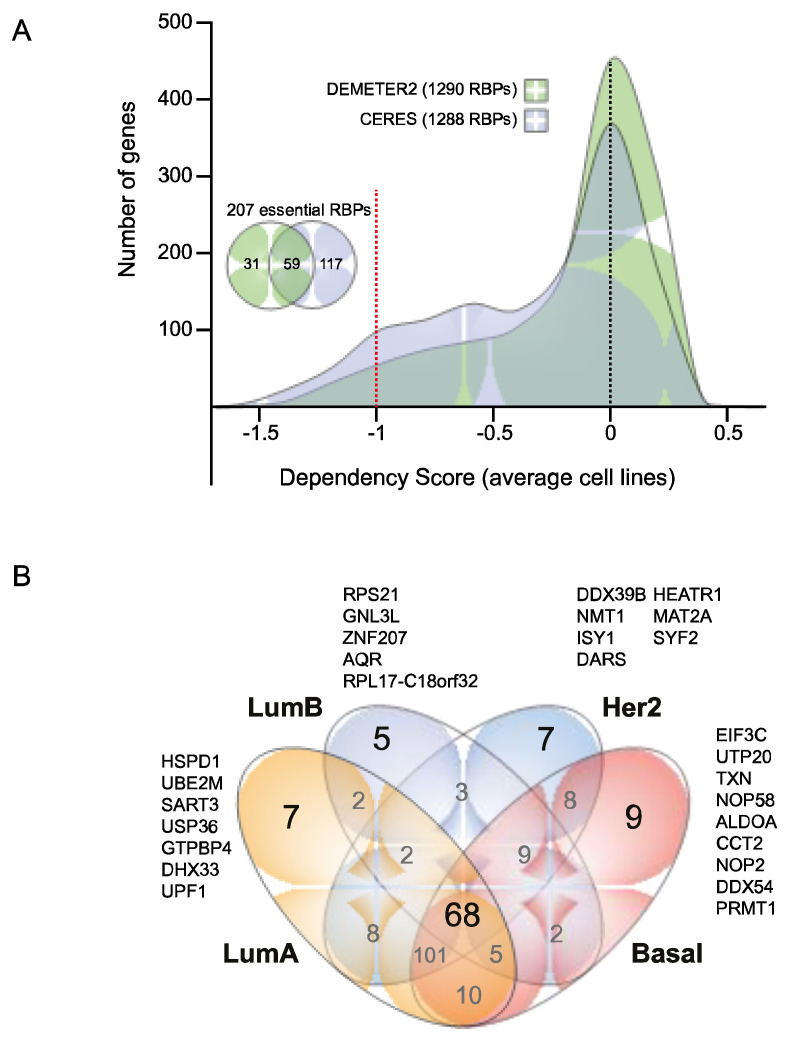
RBPs BC dependencies. (**A**) The distribution of dependency scores of 1290 (DEMETER2) and 1288 RBPs (CERES) is shown. (**B**) A Venn diagram comparing BC essential RBPs per subtype is presented.

**Figure 7 biology-11-00481-f007:**
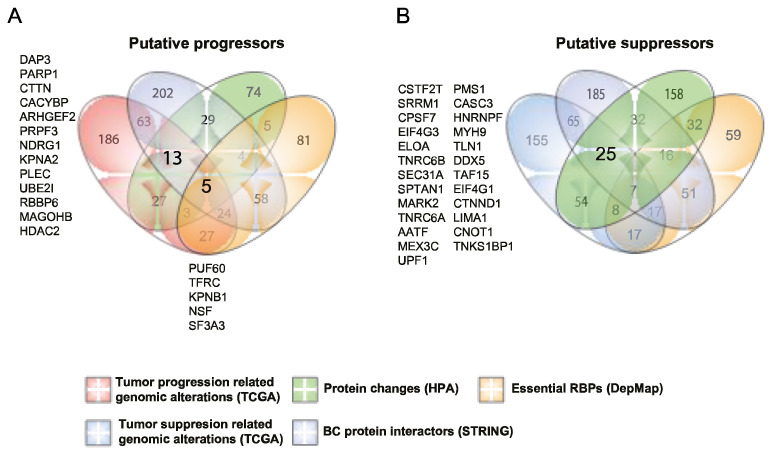
Detecting putative breast cancer RNA-binding proteins. (**A**) A Venn diagram depicting the number of unique and shared RBPs across the four cancer-progression profiles. (**B**) A Venn diagram showing the number of unique and shared RBPs across the four cancer-suppression profiles.

**Figure 8 biology-11-00481-f008:**
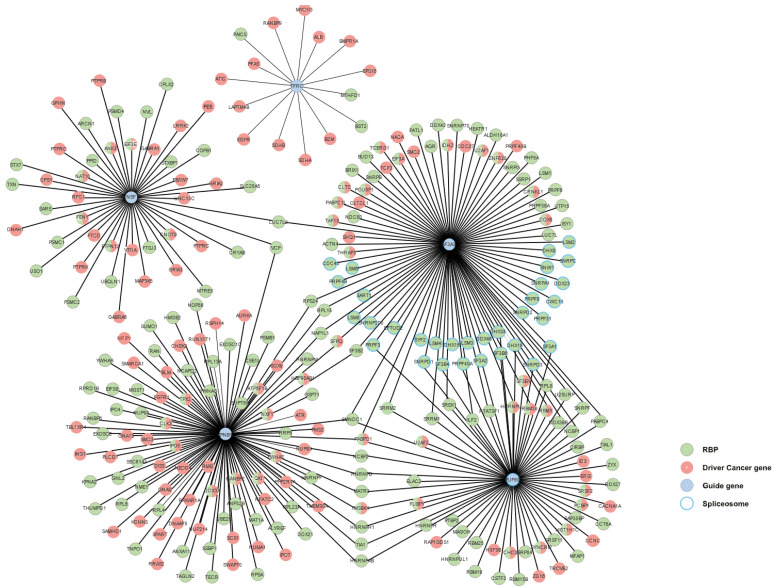
PUF60 and SF3A3 are central elements of a spliceosome-related network involving RNA-binding protein and CDGs. Previously prioritized RBPs (PUF60, TFRC, KPNB1, NSF, and SF3A3) were integrated into a disease gene network (filtered by RBPs and CDGs) using the HumanNet v2 [81,82]. Spliceosome-related proteins and their interactions were determined using MCODE [41] and CORUM [42].

**Table 1 biology-11-00481-t001:** Top ten most altered RNA-binding proteins in invasive breast carcinoma (TCGA, PanCancer Atlas) and breast cancer (CPTAC).

Genomic Alterations	Protein Name	Number of Alterations	Known BC Molecular and Cellular Functions	Related to Other Cancer Types	Pubmed Citations
Amplification + mRNA upregulation + fusion + mutations	MRPL13	579	No. However, MRPL13 is an ESR2 protein interactor in MCF7 cells [43]	No	34
	DCAF13	574	Yes. It is overexpressed in 171 primary breast tumors [44]	Yes [45]	23
	YWHAZ	532	Yes. Often amplified in BC [46], leading to increased glycolysis [47]. YWHAZ is also an ESR2 protein interactor [43]	Yes	492
	DAP3	491	Yes. DAP3 silencing contributes to breast carcinogenesis [18]	Yes [48]	77
	NUCKS1	490	Yes. NUCKS1 is overexpressed in breast tumors [49]	Yes [50]	58
	TFB2M	488	No	No	36
	MTDH	469	Yes. MTDH promotes cancer proliferation and metastasis [19]	Yes [51]	273
	C1ORF131	463	No	No	13
	PTDSS1	458	No. However, PTDSS1 is an ESR2 protein interactor in MCF7 cells [43]	No	27
	RBM34	452	No. However, RBM34 is an ESR2 protein interactor in MCF7 cells [43]	No	35
Deep deletion + mRNA downregulation + fusion + mutations	CCAR2	378	Yes. CCAR2 functions as a tumor suppressor [20]	Yes [52]	149
	DDX19A	240	No	No	24
	DHX38	180	No. However, DHX38 is an ESR2 protein interactor in MCF7 cells [43]	No	50
	ADD1	165	No. However, ADD1 is an ESR2 protein interactor in MCF7 cells [43]	Yes [53]	223
	KMT2C	135	Yes. KMT2C regulates ERα activity [54]	Yes, it is a tumor suppressor in esophageal squamous cell carcinoma [55]	88
	ZC3H18	135	No. However, ZC3H18 is an ESR2 protein interactor in MCF7 cells [43]	No	39
	NCBP3	130	No. However, NCBP3 is an ESR2 protein interactor in MCF7 cells [43]	No	26
	RARS2	123	No	No	26
	EIF4ENIF1	122	No	No	52
	NMT1	109	No	Yes [56]	92

## Data Availability

All data generated for this study are included in the manuscript and its Appendix A.

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
