# Peer review of "Integrated In Silico Analyses Identify PUF60 and SF3A3 as New Spliceosome-Related Breast Cancer RNA-Binding Proteins"

_biology, 2022, doi:10.3390/biology11040481_

Round 1

Reviewer 1 Report

García-Cárdenas et al. have utilized a robust computational biology toolkit to identify novel roles of RNA-binding proteins (RBPs) in the pathophysiology of breast cancer (BC). Their streamlined analysis pipeline of BC patient clinical datasets revealed PUF60 and SF3A3 as new BC RBPs, previously implicated in the spliceosome complex. These two players demonstrated high interconnectivity with associated cancer driver genes as well as immunohistochemical phenotypic changes in sections comparing BC against normal tissues. Furthermore, genomic alteration profiles have hinted at the tumor promoting rather than suppressing function of RBPs in BC. The study by García-Cárdenas et al. is sound and the manuscript is professionally conceived. As such, the current discovery represents an important stepping stone toward future investigations into the mechanistic facets of RBPs in orchestrating the development and progression of BC. Importantly, the work by García-Cárdenas et al. underscores the unprecedented potential of PUF60 and SF3A3 to serve as key molecular targets for future diagnosis and treatment efforts to eradicate BC. There are few minor issues that need to be addressed (see the list below).

1) Please change "Breast Cancer" to "breast cancer" (line 3).

2) Please translate "Escuela de Medicina, Facultad de Ciencias Médicas de la Salud y de la Vida, Universidad Internacional del Ecuador" into the English language (line 7).

3) Please translate "Facultade de Ciencias, Universidade da Coruña" into the English language (line 10).

4) Please translate "Instituto Nacional de Investigación en Salud Pública" into the English language (line 13).

5) Please replace "J.M.G.-C." with "J.M.G-C." (line 8).

6) Please change "A.J-A" to "A.J-A." (line 8).

7) Please replace "I.A.-C." with "I.A-C." (lines 10, 460, 462).

8) Please change "A.P.-V." to "A.P-V." (lines 10, 461, 462).

9) Please replace "A.L.-C." with "A.L-C." (lines 12, 20, 459, 460, 462).

10) Please change "The Barcelona" to "Barcelona" (line 14).

11) Please change "A.I" to "A.I." (line 15).

12) "Breast cancer (BC) is the leading cause of cancer-associated death among women worldwide" (line 21) is identical to "Breast cancer (BC) is the leading cause of cancer-associated death among women worldwide" (line 31) and very similar to "Breast cancer (BC) is the leading cause of cancer-associated death (15%: 626,679 cases) and the most commonly diagnosed cancer (24%: 2,088,849 cases) among women worldwide" (line 48). Please completely rephrase or remove two of the three sentences.

13) "aspect"/"aspects" is mentioned twice in "Breast cancer (BC) is the leading cause of cancer-associated death among women worldwide. Although several efforts have been made, several aspects remain uncertain. Thus, new aspects of breast tumorigenesis must be addressed, such as RNA-binding proteins (RBPs), which are key regulators of important cellular processes" (line 21) and on lines 32, 34, 51, 55, 63, 64, 335, 406. Please fix.

14) "Although several efforts have been made, several aspects remain uncertain" (line 22) is too vague. What the authors specifically mean by "several efforts" and "several aspects"?

15) "Although several efforts have been made, several aspects remain uncertain" (line 22) is similar in meaning to "Although large-scale efforts have completely redefined cancer, key aspects of tumor biology remain to be discovered" (line 32) and "Although these datasets have completely redefined cancer drug development, diagnosis, and treatment, additional key aspects of tumor biology remain to be discovered" (line 61). Please rephrase so that these sentences become unique in meaning.

16) "key" appears five times in Simple Summary, Abstract, and Introduction sections (lines 24, 32, 34, 63, 67) and on lines 189, 208, 335, 381. Please correct.

17) "To unravel new BC RBPs" (line 24) sounds way too similar to "To unravel new putative BC RBPs" (line 36). Please ensure that expressions in the Abstract are truly unique and don't simply repeat what has been already stated in Simple Summary.

18) "unravel"/"unraveled" appears three times in Simple Summary, Abstract, and Introduction sections (lines 24, 36, 77) and also on lines 160, 340, 418. Please fix.

19) "showing strong tumorigenic characteristics" (line 26) repeats on line 39. Please fix.

20) The meaning of "It is not unexpected that some of them have never been related to BC but possess defined tumorigenic functions in other cancer types" (line 26) overlaps with that of "Interestingly, some of these RBPs have never been studied in BC but possess defined tumorigenic functions in other cancer types" (line 41). Please rephrase so that that the meaning unambiguously differs between these two sentences.

21) In addition, "but possess defined tumorigenic functions in other cancer types" (line 27) seems to be identical to "but possess defined tumorigenic functions in other cancer types" (line 41). Please revise.

22) "defined"/"redefined" appears four times in Simple Summary and Abstract sections (lines 27, 32, 36, 41). Please rephrase.

23) "Further research should focus on the mechanisms by which these proteins could promote breast tumorigenesis, holding the promise of new therapeutic approaches along with novel drug development" (line 28) and "Further research should focus on the mechanisms by which these proteins could promote breast tumorigenesis, holding the promise of new therapeutic pathways along with novel drug development strategies" (line 43) are nearly identical. Please remove all duplicities between the Simple Summary and Abstract.

24) The information that RBPs are key post-transcriptional regulation regulators "Although large-scale efforts have completely redefined cancer, key aspects of tumor biology remain to be discovered. In that respect, post-transcriptional regulation of tumorigenesis represents an understudied aspect of cancer research. As key regulators of this process, RNA-binding proteins (RBPs) are emerging as critical modulators of tumorigenesis but only a few have defined roles in BC" (line 32) is also mentioned in "As key regulators of post-transcription, RNA binding proteins (RBPs) are particularly relevant due to their implication in every post-transcriptional step of gene expression: RNA splicing, transport, stability, translation, and localization" (line 67). Please remove this redundancy.

25) "aspects of tumor biology" (line 32) repeats on lines 55, 63, 335. Please rephrase.

26) Moreover, "In that respect, post-transcriptional regulation of tumorigenesis represents an understudied aspect of cancer research" (line 33) and "In that respect, post-transcriptional regulation of tumorigenesis represents an understudied aspect of cancer research" (line 63) are identical statements. Please ensure that each sentence is unique.

27) The meaning of "As key regulators of this process, RNA-binding proteins (RBPs) are emerging as critical modulators of tumorigenesis but only a few have defined roles in BC" (line 34) might be partially repeated in "As a result, genomic alterations of these proteins lead to dysfunctional cellular processes; indeed, RBPs are emerging as critical regulators of tumorigenesis but only a few have defined roles in BC" (line 69) and in "In this regard, RBPs are emerging as critical regulators of tumorigenesis [17]; however, they remain understudied in cancer research" (line 337).

28) It is not clear what the authors mean by "complementary bioinformatic resources" in "To unravel new putative BC RBPs, we have performed integrated in silico analyses of all human RBPs (n=1393) in three major cancer databases along with complementary bioinformatic resources" (line 36)?

29) Please change "n=1393" to "n = 1,393" (line 37).

30) Please replace "Breast" with "breast" (line 46).

31) Please change "The" to "the" (lines 55, 57, 59, 83, 153, 251).

32) Please replace "in in" with "in" (line 61).

33) Please change "regulators of post-transcription" to "post-transcriptional regulators" (line 67).

34) "To shed light on this matter, we analyzed and integrated RBPs genomic alterations, protein-protein interaction (PPI) networks, immunohistochemical profiles, and loss-of-function experiments to unravel new putative BC RBPs" (line 75) seems to be almost fully duplicated in "In the present study, we analyzed and integrated RBPs genomic alterations, PPI networks, immunohistochemical profiles, and loss-of-function experiments to unravel new putative BC proteins" (line 338). Please completely rewrite or remove one of these sentences.

35) Please replace "RBPs" with "RBP" (lines 76, 118, 151, 161, 190, 205, 247, 345, 367, 370).

36) Please replace "1393" with "1,393" (line 80).

37) Please change "1392" to "1,392" (line 83).

38) Please change "(n = 171), were" to "(n = 171) were" (line 83).

39) Please change "constructed as follows: non-cancer genes from Piazza et al. [32], without RBPs and NCG6 [31] cancer genes, were" to "constructed from Piazza et al. [32], without RBPs and NCG6 [31] cancer genes, and" (line 84).

40) The link "http://www.galseq.com/oncoscore.html" seems to be nonfunctional (line 87). Please fix.

41) Please replace "Table S1 in Supplementary File S1" with "Table S1" (lines 88, 137).

42) Please change "http://www.cbioportal.org/" to "http://www.cbioportal.org" (line 91).

43) Please change "Table S3 in Supplementary File S3" to "Table S3" (lines 97, 158).

44) Please replace "(n=1392) and BC proteins (n=171)" with "(n = 1,392) and BC proteins (n = 171)" (line 98).

45) Please change "1212" to "1,212" (lines 102, 229).

46) Please replace "https://www.proteinatlas.org/" with "https://www.proteinatlas.org" (line 104).

47) Please change "of" to "in" (line 104).

48) Please replace "Table S6 in Supplementary File S6" with "Table S6" (lines 106, 212, 230, 238, 241, 242, 244, 292, 392, 394, 397, 400).

49) Please change "1288" to "1,288" (lines 119, 255, 281, 456).

50) Please replace "1290" with "1,290" (lines 119, 256, 280, 455).

51) Please change "https://depmap.org/portal/" to "https://depmap.org/portal" (lines 120, 257).

52) Please change "Table S10 in Supplementary File S10" to "Table S10" (lines 122, 271).

53) Please replace "RBPs [n=125] and CDGs ([n=202 genes])" with "RBPs (n = 125) and CDGs (n = 202 genes)" or "RBPs, n = 125, and CDGs, n = 202 genes" (line 125).

54) Please change "https://www.inetbio.org/humannet/" to "https://www.inetbio.org/humannet" (line 127).

55) Please replace "fluff=0" with "fluff = 0" (line 129).

56) Please replace "http://mips.helmholtz-muenchen.de/corum/" with "http://mips.helmholtz-muenchen.de/corum" (line 131).

57) Please change "RBPs genomic alterations in BC." to " RNA-binding protein genomic alterations in breast cancer" (line 133).

58) Please replace "(n=1392), BC genes (n=171)" with "(n = 1,392), BC genes (n = 171)" (line 136).

59) Please change "n=170" to "n = 170" (line 136).

60) Please replace "Table S2 in Supplementary File S2" with "Table S2" (lines 142, 165, 171, 174, 353).

61) Please change "RNA-binding proteins (RBPs)" to "RBPs" (line 146).

62) Please replace "[n=170], BC genes [n=171], and RBPs [n=1392]" with "(n = 170), BC genes (n = 171), and RBPs (n = 1,392)" or ", n = 170, BC genes, n = 171, and RBPs, n = 1,392" (line 147).

63) Please change "[5–10] database" to "database [5–10]" (line 148).

64) It is not clear what the authors mean by "highly significant" (line 150)?

65) Please change "BC RBPs" to "breast cancer RNA-binding proteins" (lines 152, 202, 283, 308).

66) Please replace "RBPs: 14" with "RBPs, 14" (line 155).

67) "Tumor progressors tend to be overexpressed (mRNA upregulation or genomic amplification), while suppressors are downregulated (mRNA downregulation or genomic deletion)" (line 161) seems to lack a context. Please specify whether tumor progressors tend to be overexpressed, while tumor suppressors are downregulated, in transformed or nonmalignant cells.

68) The information that ESR2 is a tumor suppressor in breast and other cancers seems to be duplicate between "For instance, MRPL13 molecular and cellular functions have never been studied in BC; however, MRPL13 has been shown to interact with the estrogen receptor 2 (ESR2), a tumor suppressor protein in breast and other cancers" (line 166) and in "MRPL13, along with other highly altered RBPs (Table 1), has only been shown to interact with ESR2, a tumor suppressor in breast and other cancer types" (line 359).

69) Please replace "MRPL13 molecular and cellular functions" with "molecular and cellular functions of MRPL13" (line 166).

70) Despite the authors claim that "Interestingly, 30% of all human RBPs interact with the tumor suppressor ESR2 [43] (Figure 2B)" (line 169) and in "This observation led us to investigate how many RBPs interact with ESR2; strikingly, we found that 30% of all RBPs interact with this receptor" (line 361), Figure 2B shows that this number is 43.2%. Please revise.

71) Please change "[43] (Figure 2B)" to "(Figure 2B) [43]" (line 169).

72) Please change "progressors and suppressors" to "progressor and suppressor" (lines 170, 354).

73) Please replace "RBPs" with "RNA-binding proteins" (lines 175, 225, 246, 312, 326, 458).

74) Please replace "alteration s" with "alterations", "Amplificatio n" with "Amplification", "downregulat ion" with "downregulation", and "a ESR2" with "an ESR2" 7x in Table 1.

75) It might be a good idea to move "Amplificatio n + mRNA upregulation + fusion + mutations" and "Deep deletion + mRNA downregulat ion + fusion + mutations" into horizontal headings in Table 1.

76) "To further identify key RBPs" (line 187) collides with Table 1. Please fix.

77) Please change "(Normal, LumA, LumB, Her2, and Basal. Table S4 in Supplementary File S4)" to "(Normal, LumA, LumB, Her2, and Basal) (Table S4)" (line 190).

78) Please replace "(Stage I to IV; Table S5 in Supplementary File S5)" with "(Stage I to IV) (Table S5)" (line 191).

79) Please add figure reference to "Concerning staging, RBPs seem to be more altered 193 in Stage IV" (line 193).

80) From "Concerning staging, RBPs seem to be more altered in Stage IV" (line 193) is not clear to what stage are the authors comparing the RBPs to?

81) Please change "Table S4 in Supplementary File S4" to "Table S4" (line 195).

82) Please replace "Table S5 in Supplementary File S5" with "Table S5" (line 195).

83) The "B" in "Human "RBPs" is closer to "R" than to "P" in Figure 2B. Please fix.

84) Please add statistics to Figures 2C,D to highlight the differences in genomic alternation frequencies.

85) Please replace "Highly Altered" with "highly altered" (line 202).

86) Please change "the Network of Cancer Genes 6.0 (NCG6)" to "NCG6" (line 203).

87) Please change "estrogen receptor 2 (ESR2)" to "ESR2" (line 204).

88) Please replace "are also" with "are" (line 206).

89) Please change "RBPs interact with well-known BC" to "RNA-binding proteins interact with well-known breast cancer" (line 207).

90) Please replace "(n=1392) and well-known BC proteins (n=171)" with "(n = 1,392) and well-known BC proteins (n = 171)" (line 209).

91) It is not clear what the authors mean by "highest confidence" in "The interactions were obtained from experiments and databases (interaction score=0.9, highest confidence)" (line 210)?

92) Please change "score=0.9" to "score = 0.9" (line 211).

93) It is not clear what the authors refer to by "Padj" in "According to the g:Profiler [61], proteins interacting with SF3B1 are implicated in RNA splicing (Padj=3.783×10-34; GO:0000377), while proteins connected to CDC5L are mainly involved in chromatin binding (Padj=1.500×10-2; GO:0003682)" (line 214)?

94) Please replace "Padj=3.783×10-34" with "Padj = 3.783×10-34" (line 216).

95) Please format "adj" in "Padj=3.783×10-34" using subscript (line 216).

96) Please change "Padj=1.500×10-2" to "Padj = 1.500×10-2" (line 217).

97) Please format "adj" in "Padj=1.500×10-2" using subscript (line 217).

98) Please replace "PPIs between RBPs and well-known BC" with "protein-protein interactions between RNA-binding proteins and well-known breast cancer" (line 222).

99) Please change "and" to "and the" (line 223).

100) Please change "constitute" to "constitutes" (line 226).

101) Please replace "(n=130), low (n=52), medium (n=366), and high 232 (n=72)" with "(n = 130), low (n = 52), medium (n = 366), and high 232 (n = 72)" (line 232).

102) "As expected, our approach revealed well-known BC proteins, such as KPNA2 [21] or G3BP2 [22], which validate our analysis" (line 238) shares a similarity with "As expected, our approach revealed well-known BC proteins" (line 390). Please rephrase.

103) Please change "BC: DARS2" to "BC, DARS2" (line 243).

104) "presented" is mentioned twice in "(A) A correlation plot, comparing RBPs immunohistochemical levels between normal and BC tissues, is presented. Circle sizes correlate with the number of RBPs presented in each intersection" (line 247). Please fix.

105) Please replace "presented" with "present" (line 249).

106) Please change "Human Protein Atlas (HPA)" to "HPA" (line 251).

107) From the "3.5. Exploring RBPs BC dependencies" section is no clear what is a dependency score? Please briefly explain its definition to the readers in the text.

108) Please change "RBPs BC" to "breast cancer RNA-binding proteins" (line 252).

109) Please replace "Table S2-4 in Supplementary File S2-4" with "Table S2–4" (line 253).

110) Please change "RBPs BC" to "BC RBP" (lines 255).

111) Please replace "lines11" with "lines" (line 258).

112) Please replace "Table S8 in Supplementary File S8" with "Table S8" (line 264).

113) Please change "Table S9 in Supplementary File S9" to "Table S9" (line 268).

114) Please replace "Table S8-9 in Supplementary File S8-9" with "Table S8–9" (lines 269, 273, 294, 302, 408).

115) Please change "RBPs BC" to "breast cancer RNA-binding protein" (lines 280).

116) Please replace "proteins" with "proteins," (line 285).

117) Please change "BC essential" to "essential BC" (line 294).

118) Please change "characteristics:" to "characteristics," (line 296).

119) Please replace "CDGs" with "cancer driver genes" (lines 313, 326).

120) It is not clear what the authors mean by "guide genes" in "This tool allowed us to integrate these 5 RBPs (for now on guide genes) into a disease gene network" (line 315)?

121) Although the authors claim that "We first obtained an initial network of 4464 interactions (Table S11 in Supplementary File S11)" (line 317) and that "Thus, we obtained a very intricate network of 4464 interactions (Table 11 in Supplementary File S11), which emphasized the robust and complex network formed between RBP - RBP, RBP - CDG, and CDG - CDG" (line 428), only 2,231 interactions seem to be shown in Table S11. Please fix.

122) Please change "4464" to "4,464" (lines 317, 428).

123) Please change "Table S11 in 317 Supplementary File S11" to "Table S11" (lines 317, 428).

124) Please replace "(n=202 genes) and RBPs (n=125 genes)" with "(n = 202 genes) and RBPs (n = 125 genes)" (line 319).

125) Please change "biggest" to "largest" (line 320).

126) Please replace "Figure 7." with "Figure 7" (line 323).

127) Please change "Driver Cancer gene" to "Driver cancer gene" or "Cancer driver gene" in Figure 7.

128) Please remove "Authors should discuss the results and how they can be interpreted from the perspective of previous studies and of the working hypotheses. The findings and their implications should be discussed in the broadest context possible. Future research directions may also be highlighted." (line 331).

129) Please change "than" to "as" (lines 342, 376).

130) Please replace "in-silico linked" with "linked in-silico" (line 344).

131) Please change "we next" to "we" (line 349).

132) Please change "CDG: NCG6" to "CDG, NCG6" (line 350).

133) Please replace "proteins, such as DAP3 [18], MTDH [19]" with "proteins such as DAP3 [18], MTDH [19]," (line 355).

134) Please change "(MRPL13) in our analysis" to "in our analysis, MRPL13," (line 358).

135) Please add figure reference to "This observation led us to investigate how many RBPs interact with ESR2; strikingly, we found that 30% of all RBPs interact with this receptor" (line 361)

136) Please replace "implicating" with "involving" (line 364).

137) Please change "alterations accordingly" to "alterations" (line 367).

138) Please replace "Table 4-5 in Supplementary File S4-5" with "Table 4–5" (line 374).

139) Please replace "drivers: ARF1" with "drivers, ARF1" (line 377).

140) "However, CDC5L has been related to other cancer types, such as osteosarcoma [85] and prostate cancer [86], showing its possible role in breast carcinogenesis" (line 385) does not make sense since that "CDC5L has been related to other cancer types" does not necessarily indicate its "possible role in breast carcinogenesis". Please fix.

141) Please change "ex vivo [21], , is" to "ex vivo [21], is" (line 391).

142) Please replace "hepatocarcinogenesis progression" with "hepatocarcinogenesis" (line 398).

143) Please change "Worthy of note" to something like "In addition" (line 400).

144) Please replace "screens: CERES" with "screens, CERES" (line 405).

145) "research" appears twice in "Although CERES [11] and DEMETER2 [12,13] did not test all human RBPs, therapeutic posttranscriptional BC research could therefore be focused on these 59 RBPs. However, more research is needed to deeply understand their carcinogenic roles" (line 408).

146) Please change "therapeutic posttranscriptional" to "future therapeutic post-transcriptional" (lines 409, 413).

147) Please replace "processes:" with "processes such as" (line 420).

148) Please change "progressors" to "progressor" (line 426).

149) Please replace "RBP – RBP" with "RBP" (line 431).

150) Please replace "that" with "which" (line 432).

151) Please change "(e.g., splicing" to "(e.g., splicing)" (line 434).

152) Please replace "with" with "showed" or "displayed" (line 439).

153) Please replace "RBPs: PUF60" with "RBPs, PUF60" (line 441).

154) Please change "proteins" to "protein" (line 448).

155) Please replace "RBPs" with "RNA-binding" (line 454).

156) Please change "Driver Cancer Genes" to "cancer driver genes." (line 458).

157) Please change "J.M.G-C" to "J.M.G-C." (lines 459, 460 2x).

158) Please replace "S.G" with "S.G." (line 462).

159) Please change "manuscript" to "manuscript." (line 463).

160) Please ensure that the "S1" tab is active and therefore seen first in the biology-1594782-supplementary file.

161) Similarly, please ensure that the first row is immediately visible for all tabs in the biology-1594782-supplementary file.

162) It is not clear what does "msf10" refer to in Table S8 (cell H1)?

Reviewer 2 Report

This work concerns breast cancer RNA-binding proteins. It presents the results of a multistep computational study that resulted in the identification of PUF60 and SF3A3.  I have several remarks and questions:

Major remarks
1) Link http://www.galseq.com/oncoscore/ does not work for me. I get the following message "You don't have permission to access /oncoscore/ on this server." Is it a commercial tool? If not, please provide the working URL.

2) Figure 2A: do the colors of columns have any meaning? If not, I would propose to use just two colors - one for breast cancer-related columns, the other color for the other cancer types (and add a legend explaining these colors instead of this double-layer axis labeling, which is not nice).

3) "PRPF3 and MAGOHB have not been linked to cancer before" -> could you elaborate more on what were they linked to?

4) In Table 1, I cannot see NOVA1 and HNRNPK proteins. Can you explain why?

5) I suggest adding a scheme showing the workflow of the performed analysis. 

6) "Authors should discuss the results and how they can be interpreted from the perspective 331 of previous studies and of the working hypotheses. The findings and their implications 332 should be discussed in the broadest context possible. Future research directions may also 333 be highlighted." - I do not understand it. Who is the recipient of these suggestions?

7) I would like the authors to deliberate on the potential alternative binding... Is it possible to deduce this from computational exploration of the selected databases?

8) How about the relationship to the work by Lehmann, Misiewicz et al (2020)?

Minor remarks:

1) The manuscript should be reviewed for typos and errors in the text. Some of them are mentioned below:
- "not only in in healthy" -> "not only in healthy"
- "Number of alteration" -> ""Number of alterations"
- "cancer cell lines11" -> "cancer cell lines"
- "KPNA2 which has" -> "KPNA2 that has"

2) Table 1 should be in landscape orientation.
